# AUTOMATED REWARDS VIA LLM-GENERATED PROGRESS FUNCTIONS

## ABSTRACT

Large Language Models (LLMs) have the potential to automate reward engineering for Reinforcement Learning (RL) by leveraging their broad domain knowledge across various tasks. However, they often need many iterations of trial-and-error to generate effective reward functions. This process is costly because evaluating every sampled reward function requires completing the full policy optimization process for each function. In this paper, we introduce an LLM-driven reward generation framework that is able to produce state-of-the-art policies on the challenging Bi-DexHands benchmark **with 20× fewer reward function samples** than the prior state-of-the-art work. Our key insight is that we reduce the problem of generating task-specific rewards to the problem of coarsely estimating *task progress*. Our two-step solution leverages the task domain knowledge and the code synthesis abilities of LLMs to author *progress functions* that estimate task progress from a given state. Then, we use this notion of progress to discretize states, and generate count-based intrinsic rewards using the low-dimensional state space. We show that the combination of LLM-generated progress functions and count-based intrinsic rewards is essential for our performance gains, while alternatives such as generic hash-based counts or using progress directly as a reward function fall short.

## 1 INTRODUCTION

Automated reward engineering aims to reduce the human effort required when using Reinforcement Learning (RL) for sparse-reward tasks. Multiple recent efforts towards automated reward engineering have focused on leveraging Large Language Models (LLMs) to provide reward signals—either employing the LLM output directly as the reward (Kwon et al., 2023), or using the LLM to generate code for a dense reward function (Yu et al., 2023; Ma et al., 2023).

Traditionally, constructing an effective dense reward function is an intricate process of identifying key task elements and carefully weighing different reward terms (Sutton & Barto, 2018; Booth et al., 2023). Prior works on reward function generation attempt to use LLMs both for their domain knowledge and to optimize quantitative aspects of reward engineering (weighting, rescaling) (Yu et al., 2023; Ma et al., 2023)–they can require many training runs as they search through many reward functions in order to find a candidate that is effective for training (Ma et al., 2023).

Our core insight is that we can reduce the problem of reward generation to the task of generating rough measures of *task progress*. Our framework uses LLMs to generate code for *progress functions*: task-specific functions that map environment states to scalar measures of progress. For a given genre of tasks, we follow the example from prior work on reward code generation (Yu et al., 2023) by asking practitioners to provide a small helper function library (ex. *dist(x,y)*) for the particular genre's observation spaces. Then, given both the helper function library and a single-sentence description of a task within the domain (ex. *"This environment require a closed door to be opened and the door can only be pushed outward or initially open inward."*), we leverage LLMs to generate the progress functions.

We find that progress functions are most empirically effective when used within a count-based intrinsic reward framework: we treat the outputs of the progress functions as a simplified state space that groups together similar states from the environment, we discretize the states, and we compute state visitation counts across the discretized state space. Then, we treat the inverse square root of the visitation count as a count-based intrinsic reward, as in prior work (Kolter & Ng, 2009;

Tang et al., 2017), and learn policies using these count-based intrinsic rewards as task rewards. It may seem straightforward to instead directly sum the outputs of the progress function and use the sum as a reward – providing larger rewards for reaching states corresponding with greater progress through the task. However, this approach neglects the typical reward scaling and weighting issues common to dense reward shaping approaches (Sutton & Barto, 2018; Booth et al., 2023). Our progress-and-counts algorithm, ProgressCounts, achieves SOTA performance on the challenging Bi-DexHands benchmark, outperforming Eureka (Ma et al., 2023) by $4\%$. By limiting the role of LLMs to generating progress functions and applying count-based intrinsic rewards to simplified progress-based states, ProgressCounts achieves significantly greater sample efficiency compared to approaches that use LLMs directly for estimating reward weighting and scaling (Yu et al., 2023; Ma et al., 2023). Specifically, ProgressCounts requires 20 times fewer training runs than Eureka.

Specifically, we make the following contributions:

1. We re-frame the problem of reward generation in terms of generating coarse measures of *task progress*. Given a single-sentence task description and a small domain-specific library of helper functions, we leverage LLMs to generate *progress functions*.

2. We generate rewards from progress functions by treating the output of the progress function as a reduced state representation, discretizing progress in order to measure state visitation counts, and generating count-based intrinsic rewards.

3. We demonstrate that our algorithm ProgressCounts is effective by obtaining state-of-the-art performance on the Bi-DexHands benchmark. We outperform the prior state-of-the-art Eureka (Ma et al., 2023) by $4\%$ while requiring $20\times$ fewer samples.

## 2 RELATED WORK

**Automated reward engineering.** Learning directly from sparse rewards can be challenging (Ng et al., 1999; Hare, 2019; Vecerik et al., 2017). It is common for practitioners to carefully engineer dense reward functions to shape the learning process (Ng et al., 1999)—a labor-intensive (Sutton & Barto, 2018) and brittle (Booth et al., 2023) process. Advancements in automation such as Population Based Training (Jaderberg et al., 2017) have shown promise in refining this process by automating searches over fixed design spaces.

Foundation models offer the opportunity to automate reward engineering with powerful priors. The output of foundation models can be used to propose tasks for open-ended learning curricula (Du et al., 2023; Zhang et al., 2023) and add high-level structure to learning (Mirchandani et al., 2021). Foundation models can also be used directly as rewards (Fan et al., 2022; Kwon et al., 2023; Sontakke et al., 2024), and have the potential to generate reward function code, either scaffolded by reward function templates (Yu et al., 2023) or in more freeform fashion (Ma et al., 2023; Venuto et al., 2024; Li et al., 2024). We also leverage LLMs to inject task knowledge via code, but rather than attempting to generate complex reward functions, we simply ask them to identify a few key features associated with task progress.

**Count-based intrinsic rewards.** Our algorithm leverages the idea of count-based intrinsic rewards in order to convert coarse progress-based state representations into rewards for policy learning. Count-based intrinsic rewards are one of the main approaches to intrinsic motivation: estimating the "novelty" of a given state via state visitation counts (Tang et al., 2017). Approaches that hash continuous spaces to discrete representations have shown considerable promise when the discretization function is domain-specific (Tang et al., 2017; Ecoffet et al., 2021). In particular, the Go-Explore algorithm (Ecoffet et al., 2021) pairs count-based intrinsic rewards along with simulator state resets in order to achieve state-of-the-art results on several challenging Atari (Bellemare et al., 2013) tasks. The main downside to these approaches is that domain-aware discretization typically requires significant human engineering (Tang et al., 2017; Ecoffet et al., 2021). In our algorithm, by discretizing task progress, we already have access to automated domain-specific state discretizations. Absent the need for extensive human-engineered discretization functions, count-based intrinsic rewards are both practical and effective for learning policies from progress functions.

**(A) Step 1: Use LLM to Generate Task-Specific Progress Functions**

**(B) Step 2: During Training, Progress Function Guides Count-Based Reward**

Figure 1: **ProgressCounts: an algorithm for reward generation via LLM-generated task progress functions and count-based rewards.** (A) We leverage a LLM to generate code for a *progress function*, which distills task-specific features from a high-dimensional state space into a low-dimensional notion of task progress. The LLM takes as input a high-level task description, a small library of feature engineering functions, and a description of the environment state space. On a per-task basis, the user only needs to provide the task description as input. (B) We use heuristics to discretize the output of the LLM-generated progress function, compute state visitation counts across the discretized bins, and leverage standard count-based rewards to learn RL policies.

## 3 PRELIMINARIES

We consider the problem of automated reward generation for a sparse-reward task. The task is defined as a Markov Decision Process (MDP) $\mathcal{M} = (\mathcal{S}, \mathcal{A}, \mathcal{P}, \mathcal{R}, \gamma)$, where $\mathcal{S}$ is the state space, $\mathcal{A}$ is the action space, $\mathcal{P}$ is the transition probability function, and $\mathcal{R}$ is a sparse reward function that provides little guidance to the agent. The goal is to learn a policy $\pi : \mathcal{S} \to \Delta(\mathcal{A})$ that maximizes the expected cumulative reward $J(\pi) = \mathbb{E}\left[\sum_{t=0}^{\infty} \gamma^t \mathcal{R}(s_t, a_t) \middle| s_0, \pi\right]$, where $s_t, a_t$ are the state and action at time $t$.

We assume the availability of three potential inputs for reward engineering:

1. *A description of the features available in the environment* (Figure 1-A, grey). We provide a description in the form of code, similar to Ma et al. (2023); Yu et al. (2023); Singh et al. (2023)

2. *A short task description* (Figure 1-A, green), similar to Ma et al. (2023); Yu et al. (2023).

3. *An environment feature engineering library*, offering a palette of additional, higher-level features that are not task-specific but may be generally useful for solving tasks in a given domain (Figure 1-A, green). This is identical to the type of feature library in Yu et al. (2023).

Note that for (1), many learning scenarios with real-world deployment goals involve training in simulators with access to environment code (Lin et al., 2024). For (2) and (3), an experienced practitioner can quickly create a small feature engineering library, and the cost of making this library is amortized across many tasks in the same domain.

# 4 METHODS

In this section, we first introduce our algorithm for leveraging LLM domain knowledge to generate *progress functions*, which distill key features from a high-dimensional environment state space to a coarse low-dimensional notion of progress in the task. Then, we outline how we use the generated progress functions: we view progress as measure of state, and leverage count-based intrinsic rewards for learning.

## 4.1 PROGRESS FUNCTIONS

### 4.1.1 PROGRESS FUNCTION DEFINITION

Given a new task description, the first step of our process is to generate a progress function $P : \mathcal{S} \to R^k$, which takes environment features $s \in \mathcal{S}$ as input, and outputs information about the current progress of the agent on the task. Especially for more complex tasks, it may be difficult to distill a task to a single feature that tracks overall progress. Therefore, a progress function, given a state, is asked to emit a positive scalar measure of progress for one or more subtasks. For instance, for the SwingCup task (Figure 1-A), which involves 1) gripping the handles of the cup, 2) rotating the cup to the correct orientation, a good progress function would break the task into two sub-tasks and return scalars measuring progress for both sub-tasks.

Specifically, the progress function outputs $[x_1, x_2, ...x_k]$ where $x_i \in \mathcal{R}$ tracks task progress for sub-task $i$. It also outputs additional variables $[y_1, y_2, ...y_k]$ that inform our framework whether the progress variables $x_i$ are increasing or decreasing.

### 4.1.2 PROGRESS FUNCTION GENERATION

For any given task, domain knowledge is required in order to determine what features from the environment are useful for assessing progress, and how to compute progress from those features. We derive that domain knowledge from an LLM, which is used to generate code for the progress function $P$. In order to translate domain knowledge into an effective progress function, we provide the LLM with three inputs:

1. **Function inputs**: we specify the features available as inputs to the progress function via a description of the features in the environment state (Figure 1-A, grey). This information is available via the simulator.

2. **Function outputs**: we specify the desired output of the progress function via a short task description (Figure 1-A, green). Humans specify this information on a per-task basis.

3. **Function logic**: we structure the process of translating feature inputs to progress output by providing the LLM with access to a *environment feature engineering library* (Figure 1-A, green). This library offers a palette of additional, higher-level features to optionally use to compute progress, and also indirectly suggests that certain types of feature transformations are beneficial to compute progress (ex. $l_2 dist(x, tgt)$). Humans create this library once per genre or benchmark of tasks.

Note that on a per-task basis, ProgressCounts only requires a user to provide the short task description. Please see Appendix A.3.2 for the libraries used for our Section 5 benchmarks.

Given the high-level task description, the small feature engineering library, and code describing the environment state space, we follow standard LLM prompting strategies to generate the progress function code (Figure 1-A), blue). Please see Appendix A.4 for several examples of generated progress functions.

## 4.2 FROM PROGRESS TO REWARD

Given a perfect estimate of task progress, it may seem natural to directly use progress as a dense reward: for a given state $s$, compute the sum of the progress outputs $\mathcal{R}_{sum} = \sum_i x_i$, and use progress sum $\mathcal{R}_{sum}$ as the reward for reaching $s$. However, learning from dense rewards can be brittle–small mistakes in reward design can often lead to a failure to learn effective policies (Booth et al., 2023). Progress functions offer highly simplified state representations—and given the coarse nature of these representations, we look for a more forgiving mechanism to generate rewards from these simplified representations. Therefore, we use a count-based intrinsic reward approach inspired by prior work that achieves state-of-the-art performance with domain-specific discretizations (Ecoffet et al., 2021).

### 4.2.1 PRELIMINARIES: COUNT-BASED REWARDS

To facilitate exploration, we leverage count-based rewards proportional to state novelty (Kolter & Ng, 2009): $n(s) \propto \frac{1}{\sqrt{c(s)}}$, where $c(s)$ is the state visitation count. High-dimensional state spaces require a binning function $B : \mathcal{S} \to \mathcal{S}'$ that maps state space $\mathcal{S}$ to a (small) discrete space $\mathcal{S}'$ where we can tractably compute state visitation frequencies and novelty $n(s) \propto \frac{1}{\sqrt{c(B(s))}}$.

When learning a policy, sparse extrinsic rewards are augmented with a standard intrinsic reward proportional to $n(s)$ (Tang et al., 2017):

$$\mathcal{R}_{total}(s_t, a_t) = \mathcal{R}(s_t, a_t) + \lambda_c n(B(s_{t+1})) \tag{1}$$

Where $s_t$ is the state at time $t$, $a_t$ is the action, $s_{t+1}$ is the next state, and $\lambda_c$ is a hyperparameter weighting the intrinsic reward relative to extrinsic reward $\mathcal{R}(s_t, a_t)$.

Effective state binning should encode information if and only if it is relevant to solving the particular task (Tang et al., 2017; Ecoffet et al., 2021).

### 4.2.2 COUNT-BASED REWARDS FROM PROGRESS

We automatically generate a task-specific binning function $B$ by converting the continuous progress values $P(s)$ emitted by the progress function to discrete states using a mapping $D : R^k \to S'$ that:

1. Estimates relevant value ranges $(min_i, max_i)$ for each $x_i$ from progress data
2. Discretizes within $(min_i, max_i)$ to produce discrete progress features $x_i'$. We discretize later subtasks with finer granularity in order to encourage more exploration closer to the goal.
3. Defines $B(s) = D(P(s)) = \sum_i x_i'$.

Heuristic discretization avoids the need to learn scalar progress ranges from environment interaction, an approach used in prior work (Shinn et al., 2024; Ma et al., 2023). While we could have leveraged the LLM directly to emit logic for discretizing progress features, we chose to use these heuristics instead since LLMs are known to struggle with numerical reasoning (Shen et al., 2023). Details and example discretization code are included in Appendix A.6.

Having defined mapping $B$ to discretize progress features, we measure state novelty from bin visitation counts via $n(s) \propto \frac{1}{\sqrt{c(B(s))}}$, augmenting existing sparse extrinsic rewards. As shown in Figure 1-B, we learn policies using Proximal Policy Optimization (PPO) (Schulman et al., 2017), augmenting the sparse extrinsic task rewards with intrinsic rewards via count-based intrinsic motivation (See Eq. 1).

## 5 EVALUATION

We evaluate ProgressCounts by using it to train policies on Bi-DexHands: a challenging sparse-reward benchmark consisting of 20 bimanual manipulation tasks. We also include additional results from the MiniGrid benchmark in the Appendix.

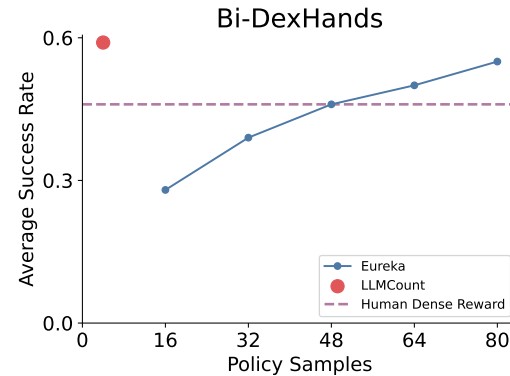

Figure 2: **On the Bi-DexHands benchmark, ProgressCounts produces policies that perform comparably to those of Eureka in terms of average task success rate, at a much smaller sample budget.** Eureka's evolutionary algorithm requires 48 policy samples (training runs with different generated reward functions) to find a policy whose performance matches that of human-designed dense reward functions. ProgressCounts requires only four policy samples (different progress functions), generating a policy that outperforms the human-designed baseline and exceeding the peak performance achieved by Eureka after 80 policy samples (20× the cost of ProgressCounts).

### 5.1 EXPERIMENTAL SETUP

**Bi-DexHands.** The Bi-Dexterous Manipulation benchmark (Chen et al., 2022) (Bi-DexHands) consists of 20 bimanual manipulation tasks with continuous state and action spaces, such as using two robotic hands to lift a pot or simultaneously pass objects between the hands. These tasks have sparse rewards and require complex coordinated motion, making them a challenging test for leveraging language models to guide policy learning. Following conventions from prior work (Ma et al., 2023), progress functions have acess to the environment state space code, and we evaluate performance on Bi-DexHands in terms of the policy's success rate at completing each task, averaged over five trials (policy training runs with different seeds).

We use Bi-DexHands to evaluate the policy performance and sample efficiency of ProgressCounts against three baselines. 1) **Sparse** extrinsic rewards upon task success, 2) **Dense**: expert-written dense extrinsic rewards from the original benchmark, 3) rewards generated using the **Eureka** LLM-based reward generation algorithm (Ma et al., 2023), the current state-of-the-art reward generation method on Bi-DexHands.

**Training configuration.** In all experiments we train policies using PPO (Schulman et al., 2017). We train policies using the PPO hyperparameters and sample budgets (100M environment samples) established by Bi-DexHands Chen et al. (2022), also used in prior work Eureka Ma et al. (2023). We set the intrinsic reward coefficient $\lambda_c = 1e - 3$, and discretize progress into 1000 bins. We leverage GPT-4-Turbo ('gpt-4-turbo-2024-04-09') as the LLM (Achiam et al., 2023) used to generate progress functions. Following the experimental procedure from prior work (Ma et al., 2023), **we use the LLM to generate multiple options for the progress function, and select the resulting policy that achieves the highest success from a single training run**–we refer to the different trained policies as *policy samples*. Unless otherwise specified, ProgressCounts uses four policy samples per task, and all policies are trained using 100M environment samples (number of environments × number of simulation steps). All LLM prompts, including task descriptions and environment feature engineering primitives, are included in Appendix A.3.

### 5.2 COMPARISON TO EXTRINSIC REWARD BASELINES

**ProgressCounts trains policies that (on average) outperform those from Eureka on Bi-DexHands, using only 5% of Eureka's training budget.** Averaged over all Bi-DexHands tasks, ProgressCounts achieves a success rate of 0.59, 13% higher than human-written dense rewards, and 4% higher than Eureka, the state-of-the-art method on this benchmark (Figure 2). Most importantly,

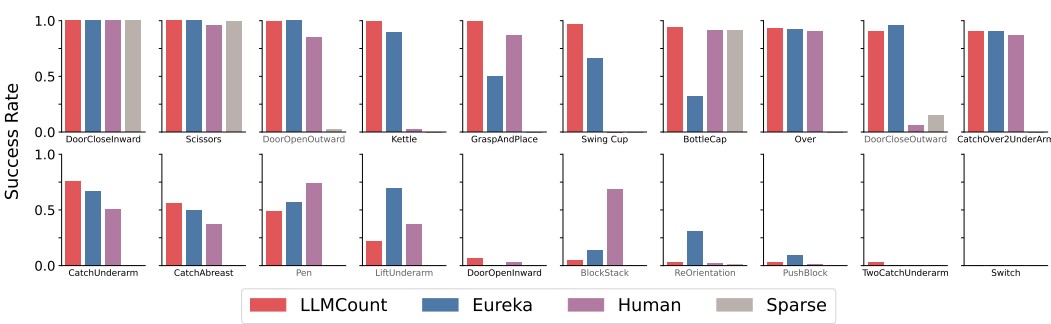

Figure 3: **ProgressCounts produces policies whose performance (in terms of task success rate) matches or exceeds that of the prior state-of-the-art method (Eureka) on 13 of 20 tasks in the Bi-DexHands benchmark.** Sparse rewards (Sparse) struggle to learn effective policies for most Bi-DexHands tasks. See Appendix A.7 for results in tabular form.

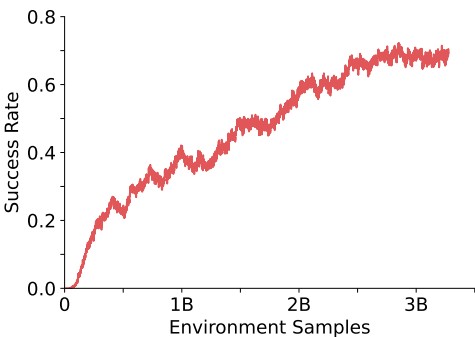

Figure 4: By allocating many environment samples to a single training run, ProgressCounts trains a policy that achieved high success on the challenging TwoCatchUnderarm task. All baselines achieved zero success on this task given a two billion environment sample budget.

Eureka's evolutionary algorithm requires 80 policy samples (generated reward functions) to find good reward functions for the task. In contrast, ProgressCounts only requires four policy samples (generated progress functions). By structuring reward engineering around a constrained progress function and heuristic discretization, we reduce the unreliability associated with using LLMs for unconstrained code generation Yu et al. (2023). This approach also allows for more robust state discretization for count-based intrinsic rewards, using only a limited number of progress function generation attempts, *without requiring costly feedback-driven evolution*.

Figure 3 presents policy performance for all 20 tasks in the Bi-DexHands benchmark. Across the benchmark, ProgressCounts matches or exceeds Eureka in performance on 13 of the 20 tasks, and ProgressCounts matches or exceeds the performance of the expert-written dense reward (Human) on 17 of the 20 tasks.

**Given the same environment sample budget as Eureka, ProgressCounts can produce higher-performance policies.** Since ProgressCounts requires fewer policy samples to find good policies, users can more confidently allocate significant fractions of a training budget to a small number of policy samples. We use ProgressCounts to train four policies on the TwoCatchUnderarm task, each for a total of two billion environment samples (the same number of samples used in aggregate, across all policy samples, for Eureka training). The best resulting policy achieves a task success rate of 0.55, and continues to improve with further samples (Figure 4). On the other hand, all extrinsic reward baselines, as well as ProgressCounts trained on 400 million environment samples (four policy samples, with 100 million environment samples each), achieve a success rate of nearly zero on this

task (Figure 3). To our knowledge, ProgressCounts is the first method to achieve reasonable success on this challenging task.

## 5.3 METHOD ABLATIONS

| Task name | ProgressCounts | ProgressAsReward | SimHashCounts |
|---|---|---|---|
| Average | **0.59** | 0.45 | 0.34 |
| Over | **0.93** | 0.90 | 0.91 |
| DoorCloseInward | **1.00** | **1.00** | **1.00** |
| DoorCloseOutward | 0.90 | **1.00** | 0.76 |
| DoorOpenInward | **0.07** | 0.00 | 0.00 |
| DoorOpenOutward | **0.99** | 0.31 | **0.99** |
| Scissors | **1.00** | **1.00** | **1.00** |
| SwingCup | 0.97 | **0.99** | 0.94 |
| Switch | **0.00** | **0.00** | **0.00** |
| Kettle | **0.83** | 0.00 | 0.00 |
| LiftUnderarm | **0.22** | 0.08 | 0.00 |
| Pen | **0.49** | 0.22 | 0.09 |
| BottleCap | **0.94** | 0.04 | **0.94** |
| CatchAbreast | **0.56** | 0.49 | 0.00 |
| CatchOver2UnderArm | 0.90 | **0.94** | 0.00 |
| CatchUnderarm | 0.76 | **0.88** | 0.00 |
| ReOrientation | 0.03 | **0.06** | 0.02 |
| GraspAndPlace | **0.99** | 0.98 | 0.08 |
| BlockStack | 0.05 | 0.00 | **0.06** |
| PushBlock | **0.03** | 0.02 | 0.01 |
| TwoCatchUnderarm | **0.03** | 0.01 | 0.00 |

Table 1: **An ablation testing whether Progress Functions and Count-Based Rewards are both necessary for ProgressCounts across the 20 tasks in Bi-DexHands. ProgressCounts** is our algorithm. **ProgressAsReward** takes the best generated progress functions, and directly uses the summed progress variables as a dense reward function. **SimHashCounts** applies SimHash to the observation space as the binning function instead of progress-based bins (the method from [4]). Results are averaged across 5 trials for **ProgressCounts**, and are single-trial numbers for the ablated methods. **ProgressCounts requires both key components of the algorithm for an** 0.59 **average task success rate.**

The success of ProgressCounts is due to both the use of progress functions to collapse simulator states into bins and due to the effectiveness of count-based intrinsic exploration applied to these bins. While progress function might seem a suitable dense reward, progress-based rewards only achieve a success rate of 0.45 (Table 1), so best performance is achieved when using count-based exploration across discretized progress bins. Both LLM-generated progress functions and count-based intrinsic exploration are necessary to achieve our SOTA performance.

**Progress functions generate more effective bins than SimHash:** On Bi-DexHands, Table 1 highlights that ProgressCounts achieves a success rate of 0.59 with progress-based bins, and only achieves a success rate of 0.34 with SimHash-based (Sadowski & Levin, 2007) bins across the observation space (Tang et al., 2017). Across the benchmark, progress-based bins achieve performance equal to or better than SimHash-based bins across 19 of 20 tasks, with the remaining task (BlockStack) within the margin of error (see Table 8 in the Appendix for standard deviations). This result aligns with prior work on human-written hash functions for count-based rewards, where the integration of domain knowledge consistently improves performance (Tang et al., 2017; Ecoffet et al., 2021).

**Count-based rewards are more effective than directly using progress as reward** While the sum of the outputs of a progress function $\mathcal{R}_{sum} = \sum_i x_i$ might seem a viable reward signal for learning, Table 1 illustrates that progress-based dense rewards only achieve a success rate of 0.45,

| Task | Default | No feature library | No heuristic discretization |
|---|---|---|---|
| SwingCup | 0.97 | 0.90 | 0.00 |
| CatchUnderarm | 0.76 | 0.00 | 0.76 |
| DoorCloseOutward | 0.90 | 0.86 | 0.92 |

Table 2: **Both the environment feature library and heuristic progress discretization help task success rate.** When the feature engineering library is removed from the LLM prompt, we obtain comparable performance on SwingCup and DoorCloseOutward, but the CatchUnderarm policy completely fails to learn. When we ask the LLM to directly generate code for discrete bins (removing heuristic discretization), SwingCup fails to learn an effective policy.

while progress functions paired with count-based intrinsic rewards achieve a success rate of 0.59, matching or outperforming the dense-reward alternative on 15 of 20 tasks. This result highlights that, given coarse state representations from progress functions, count-based intrinsic rewards are more effective than standard dense rewards.

**The environment feature engineering library helps generate effective progress features**    In Table 2, we ablate the impact of providing a feature engineering library to help ProgressCounts with generating code to compute progress features. When we remove the feature library, we still obtain comparable performance on both SwingCup (task success rate of 0.97 vs. 0.90) and DoorCloseOutward (0.90 vs. 0.86), but the CatchUnderarm policy fails to learn completely. Upon inspection of the generated code in Appendix A.5, the LLM chooses to incorporate object linear velocity into the progress function, a variable that is not directly relevant to task success. This variable is averaged with more relevant variables to derive an overall progress metric. Having incorrectly modeled task progress, the policy's unnecessary exploration is likely responsible for task failure. The library for Bi-DexHands (included in Appendix A.5) only contains functions measuring Euclidean and rotational distance; we hypothesize that knowledge of transformations available in the feature library helps the LLM ignore features for which the library is not applicable (ex. the irrelevant velocity features).

**ProgressCounts benefits from using heuristics to discretize progress features**    In Table 2, we also ablate the impact of using heuristics to discretize and combine progress features–instead, we ask the LLM to directly generate code to output discrete bins corresponding to task progress. We obtain comparable task success rate on DoorCloseOutward (0.90 vs. 0.92) and CatchUnderarm (0.76 vs. 0.76), but the trained policies completely fail without heuristic discretization for SwingCup— as seen in Appendix A.5, the LLM incorrectly guesses the relevant range of values for multiple features, and as a result the binning function is ineffective for facilitating task-relevant exploration. Heuristic discretization helps avert this failure mode when constructing task-specific state binning functions.

## 6 DISCUSSION

The state-of-the-art results achieved by ProgressCounts demonstrate two key takeaways:

First, LLM-generated progress functions offer a compelling mechanism to generate coarse task-specific state representations, and alongside count-based intrinsic rewards offer an empirically-superior alternative to using LLMs to engineer reward functions. ProgressCounts outperforms Eureka, which uses LLMs to engineer reward functions, both in terms of performance and sample efficiency. One reason for this success is the structure (task progress, count-based rewards, etc.) we build into the ProgressCounts framework, which increases the quality and reliability of LLM responses, and reduces the need for trial and error across reward weights and scaling. Perhaps more interestingly, we hypothesize that ProgressCounts also benefits from count-based intrinsic rewards being robust to non-optimal binning functions, unlike reward functions where even minor errors can easily lead to a failure to solve tasks successfully.

Second, despite being relatively under-utilized in recent research, count-based intrinsic rewards can be surprisingly effective at training policies that operate in complex high-dimensional state spaces *when given an adequate binning function*. Interestingly, the results achieved by ProgressCounts

suggest that these binning functions do not need to be hugely complex; ProgressCounts outperforms state-of-the-art, human-engineered dense reward functions using count-based exploration driven by binning functions that contain less than 20 lines of code.

Overall, we believe ProgressCounts represents a novel and promising strategy for injecting domain knowledge from large language models into an RL training loop. We hope that these results will encourage further research into (and more general usage of) count-based intrinsic methods, as well as exploration of other novel methods for leveraging LLMs to assist in solving reinforcement learning tasks.

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

# A APPENDIX

## A.1 ABLATING THE CHOICE OF CONSTANTS FOR PROGRESSCOUNTS

ProgressCounts does not require search over hyperparameters on a per-task basis–we set the hyperparameters once per benchmark and did not tune them (following the same experimental protocol as Eureka (Ma et al., 2023)). We evaluate ProgressCounts 's robustness to different hyperparameters using two ablations: 1) the intrinsic reward weight $\lambda_c$, and 2) the number of discrete bins used, each tested across three tasks from Bi-DexHands. Table 3 shows similar task performance for $\lambda_c$ of $1e-2$, $1e-3$, and $1e-4$, with the exception of slightly lower task performance on CatchUnderarm for 1e-4. Table 4 shows similar task performance with 500, 1000, and 2000 bins across the three tasks. There are no clear trends in performance across parameters in either table, and our chosen parameters ($\lambda_c = 1e-3$, 1000 bins) are actually sub-optimal for most tasks.

| Task | $\lambda_c = 0.01$ | $\lambda_c = 0.001$ | $\lambda_c = 0.0001$ |
|------|------|------|------|
| SwingCup | 0.95 | 0.97 | 0.98 |
| CatchUnderarm | 0.8 | 0.76 | 0.4 |
| DoorCloseOutward | 0.99 | 0.9 | 0.85 |

Table 3: **Ablating the impact of intrinsic reward coefficient $\lambda_c$ on ProgressCounts performance.** We report success rates for 3 values across 3 Bi-DexHands tasks. Success rates are averaged across 5 trials. Default is $\lambda_c = 0.001$

| Task | 500 bins | 1000 bins | 2000 bins |
|------|------|------|------|
| SwingCup | 0.97 | 0.97 | 0.97 |
| CatchUnderarm | 0.78 | 0.76 | 0.67 |
| DoorCloseOutward | 0.84 | 0.9 | 0.99 |

Table 4: **Ablating the impact of the number of bins on ProgressCounts performance.** We report success rates for 3 values across 3 Bi-DexHands tasks. Success rates are averaged across 5 trials. Default is 1000 bins.

## A.2 MINIGRID EXPERIMENTS

Having demonstrated that ProgressCounts yields state-of-the-art performance on the Bi-DexHands benchmark, we further evaluate how well the progress-based counts from ProgressCounts can serve as a novelty metric within more sophisticated intrinsic motivation algorithms. Specifically, we test how well progress-based novelty from Section 4.2 performs as a novelty measure within the NovelD meta-criterion (Zhang et al., 2021), which provides reward proportional to the difference in novelty between consecutive states:

$$\mathcal{R}_{total}(s_t, a_t) = \mathcal{R}(s_t, a_t) + \lambda_c \max(n(B(s_{t+1})) - \alpha n(B(s_t)), 0) \mathbb{1}[n_e(s_{t+1}) = 1] \quad (2)$$

Where $\alpha \in [0, 1]$ discounts previous novelty, and $n_e(s_{t+1})$ measures episodic novelty, measuring the number of visits to a state within the current episode.

Using the MiniGrid benchmark (Chevalier-Boisvert et al., 2024), we evaluate on a subset of eight difficult exploration tasks across two task distributions: four KeyCorridor variants and four ObstructedMaze variants. These tasks provide sparse rewards upon task success, and these rewards are proportional to the efficiency of completing the goal. We compare the efficacy of the NovelD exploration meta-criterion (Zhang et al., 2021) when measuring novelty with **ProgressCounts** as well as **RND** Burda et al. (2018), the method used in the original NovelD paper. Following Zhang et al. (2021), we measure episode rewards averaged over four trials when combining the novelty metrics with the NovelD algorithm–we measure performance in terms of the samples required to reach a threshold task reward. We set $\lambda_c = 0.1$, and we do not require progress discretization on this

| Env Type | Layout | ProgressCounts | RND |
|---|---|---|---|
| **KeyCorridor (Medium)** | S3R3 | 0.2 | 0.5 |
| | S4R3 | 0.5 | 0.9 |
| | S5R3 | 0.9 | 1.3 |
| | S6R3 | 1.2 | 1.7 |
| **ObstructedMaze (Hard)** | 2Dlhb | 1.6 | 4.0 |
| | 1Q | 1.0 | 2.8 |
| | 2Q | 2.5 | 4.7 |
| | Full | 2.9 | 7.2 |

Table 5: **ProgressCounts is more sample-efficient than RND when used as a novelty metric within NovelD**. We measure the number of environment samples ($\times 10^7$) required for NovelD to pass a threshold reward of 0.75 (except for ObstructedMaze-Full, where computational constraints limit us to a threshold of 0.5). Across four variants of the KeyCorridor task and four variants of the ObstructedMaze task, ProgressCounts is up to $64\%$ more sample-efficient than RND.

environment since the progress functions are already discrete. As with the Bi-DexHands experiments, we run 4 trials of progress function generation.

Table 5 compares the performance of ProgressCounts and RND as a novelty metric within NovelD on MiniGrid. Across both KeyCorridor and ObstructedMaze families of tasks, ProgressCounts improves the sample efficiency of NovelD at reaching a threshold reward compared to RND. This trend holds across progressively more complicated tasks: ProgressCounts improves sample efficiency by $60\%$ for KeyCorridorS3R3, the simplest task, and also improves sample efficiency by $60\%$ on ObstructedMaze-Full, the hardest task. Note that ProgressCounts is also more computationally efficient than RND, which learns an additional network to output intrinsic rewards (Burda et al., 2018). Full training curves are in Appendix A.9.

## A.3 DETAILS ON LLM INPUTS

### A.3.1 SYSTEM PROMPT

```
1  You are a reinforcement learning engineer trying to write progress
   ↪  functions to solve reinforcement learning tasks as effectively
   ↪  as possible.
2  Your goal is to identify the variables for the environment that are
   ↪  maximally relevant for measuring progress in the task described
   ↪  in text.
3  Some tasks may have only a single stage, and some tasks may have two
   ↪  separate stages.
4  You will be provided with a definition of the observation space for
   ↪  a reinforcement learning environment, and also provided with a
   ↪  small set of helper functions that can be used to transform the
   ↪  variables in the observation space.
5  Write a function that returns the variable most associated with task
   ↪  progress for each stage of the task.
6  This function can take as input any member of self defined in
   ↪  compute_observations, and can apply any of the helper functions
   ↪  to any variables from self.obs_buf to generate new derived
   ↪  features (ex. computing the distance between object and goal).
   ↪  If a single stage requires multiple progress variables, average
   ↪  the variables.
7  Also return a bool for each variable that is True if progress
   ↪  requires the variable to increase, and False if it requires the
   ↪  variable to decrease.
8
9  Function signature:
```

```
10  def progress_function(self) -> Tuple[List[torch.Tensor],
    ↪   List[bool]]:
11      # Logic here
12      return progress_vars, progress_directions
```

### A.3.2 ENVIRONMENT FEATURE ENGINEERING LIBRARY

**Bi-DexHands**   For the Bi-DexHands benchmark, our library is composed of three simple functions:
1) Euclidean distance, 2) rotational distance between quaternions, 3) Euclidean distance to a 'goal'
state if it exists.

```
1   # Determine distance of an object from a "goal", if it exists
2   def goal_dist(self, x):
3       return torch.norm(self.goal_pos - x, p=2, dim=-1)
4
5   # Determine distance between two objects
6   def dist(self, x, y):
7       return torch.norm(x - y, p=2, dim=-1)
8
9   # Rotational distance
10  def rot_dist(self, object_rot, target_rot):
11      quat_diff = quat_mul(object_rot, quat_conjugate(target_rot))
12      rot_dist = 2.0 * torch.asin(torch.clamp(torch.norm(quat_diff[:,
        ↪   0:3], p=2, dim=-1), max=1.0))
13      return rot_dist
```

**MiniGrid**   For the MiniGrid benchmark, our library is composed of three simple functions: 1)
breadth-first search to find the shortest path between two grid cells, accounting for walls, 2) a function
finding the grid position of a given type of object, 3) a function that finds the grid position of an
object, given that the object is on the path between two given locations.

```
1   def bfs(grid, start, end):
2       """
3       Perform BFS to find the shortest path from start to end in a
        ↪   minigrid environment.
4       Args:
5       - grid (np.array): The grid represented as a numpy array of
        ↪   shape (n, m, 3).
6       - start (tuple): Starting position (x, y).
7       - end (tuple): Ending position (x, y).
8
9       Returns:
10      - path (list): List of tuples as coordinates for the shortest
        ↪   path, including start and end.
11                   Returns an empty list if no path is found.
12      """
13      queue = deque([start])
14      paths = {start: [start]}
15      directions = [(1, 0), (0, 1), (-1, 0), (0, -1)]  # Down, right,
        ↪   up, left
16      while queue:
17          current = queue.popleft()
18          #if grid[current[0], current[1], 0] != 1:
19          #    print("Current", current)#, paths[current])
20          #    print("Content", grid[current[0], current[1]])
21          if current == end:
22              return paths[current]
23          for direction in directions:
24              neighbor = (current[0] + direction[0], current[1] +
                ↪   direction[1])
25              if (0 <= neighbor[0] < grid.shape[0] and
```

```
26                      0 <= neighbor[1] < grid.shape[1] and
27                      neighbor not in paths and
28                      is_traversable(grid[neighbor])):
29                      paths[neighbor] = paths[current] + [neighbor]
30                      queue.append(neighbor)
31      return []   # Return an empty list if no path is found
32
33  def get_position(grid, object_type, color=None):
34      """
35      Get the position of the object of the specified type in the
        ↪  grid.
36      Args:
37      - grid (np.array): The grid represented as a numpy array of
        ↪  shape (n, m, 3).
38      - object_type (int): The type of the object to find.
39
40      Returns:
41      - position (tuple): The position of the object in the grid.
42      """
43      for i in range(grid.shape[0]):
44          for j in range(grid.shape[1]):
45              if grid[i, j, 0] == object_type:
46                  if color is not None and grid[i, j, 1] != color:
47                      continue
48                  return (i, j)
49      return None
50
51  def get_position_on_path(grid, agent_pos, final_pos, object_type,
    ↪  color=None, closed=None):
52      path = bfs(grid, agent_pos, final_pos)
53      for pos in path:
54          if grid[pos[0], pos[1], 0] == object_type:
55              if color is not None and grid[pos[0], pos[1], 1] !=
                ↪  color:
56                  continue
57              if closed is not None and grid[pos[0], pos[1], 2] !=
                ↪  closed:
58                  continue
59              return pos
60      return None
```

### A.3.3 TASK DESCRIPTIONS

**Bi-DexHands Environments**
Task name
Task description
Task success condition

**Over**
This environment requires an object in one hand to be thrown to the goal location on the other hand. The task is a simple single-stage task.
$1[dist < 0.03]$

**DoorCloseInward**
This environment require a closed door to be opened and the door can only be pushed outward or initially open inward.
$1[door\_handle\_dist < 0.5]$

**DoorCloseOutward**
This environment requires a closed door to be opened, but because they can't complete the task by simply pushing, we need to catch the handle by hand and then open it, so it is relatively difficult.
$1[door\_handle\_dist < 0.5]$

**DoorOpenInward**

This environment requires the hands to grab the handles of the doors, then pull the two doors apart.

$1[door\_handle\_dist > 0.5]$

**DoorOpenOutward**

This environment requires the hands to grab the handles of the doors, then pull the two doors apart.

$1[door\_handle\_dist < 0.5]$

**Scissors**

This environment requires the hands to grab the handles of a pair of scissors, then open the scissors.

$1[dof\_pos > -0.3]$

**SwingCup**

This environment involves two hands and a dual handle cup, we need to use two hands to hold and swing the cup to a target orientation.

$1[rot\_dist < 0.785]$

**Switch**

This environment requires both hands to reach their respective switches, then lower the switch handle positions by applying a strong downward force.

$1[1.4 - (left\_switch\_z + right\_switch\_z) > 0.05]$

**Kettle**

This environment requires the hands to grab the kettle, then move the kettle spout to the bucket.

$1[bucket - kettle\_spout| < 0.05]$

**LiftUnderarm**

This environment requires grasping the pot handle with two hands and lifting the pot to the designated position.

$1[dist < 0.05]$

**Pen**

This environment requires the cap to be removed from the pen.

$1[5 \times |pen\_cap - pen\_body| > 1.5]$

**BottleCap**

This environment involves two hands and a dual handle cup, we need to move the cap away from the bottle.

$1[dist > 0.03]$

**CatchAbreast**

This environment consists of two shadow hands placed side by side in the same direction and an object that needs to be passed from one palm to a goal position on the other.

$1[dist < 0.03]$

**CatchOver2Underarm**

This environment requires an object in one hand to be thrown to the goal location on the other hand.

$1[dist < 0.03]$

**CatchUnderarm**

This environment requires an object in one hand to be thrown to the goal location on the other hand.

$1[dist < 0.03]$

**ReOrientation**

This environment involves two hands and two objects. Each hand holds an object and we need to reorient the object to the target orientation.

$1[rot\_dist < 0.1]$

**GraspAndPlace**

This environment consists of dual-hands, an object and a bucket that requires us to pick up the object and put it into the bucket.

$1[block - bucket| < 0.2]$

**BlockStack**

This environment involves dual hands and two blocks, and we need to stack the block as a tower.

$1[goal\_dist\_1 < 0.07 \; and \; goal\_dist\_2 < 0.07 \; and \; 50 \times (0.05 - z\_dist\_1) > 1]$

**PushBlock**

This environment involves dual hands and two blocks, and we need to push both blocks to goal positions.

$1[0.1 \leq left\_dist \leq 0.1 \; and \; right\_dist \leq 0.1]$

$1[0.5 \times |left\_dist - 0.1 \; and \; right\_dist \leq 0.1]$

**TwoCatchUnderarm**

This environment requires two objects to be thrown into the other hand at the same time.

$1[dist < 0.03]$

**MiniGrid Environments**
Task name
Task description
Sparse reward upon episode end

**KeyCorridor**

This environment is a Gridworld that requires the agent to navigate to a key, then to a blue ball.

A reward of $1 - 0.9 * (step\_count/max\_steps)$ is given for success, and 0 for failure.

**ObstructedMaze**

This environment is a Gridworld that requires the agent to navigate to a blue ball, and there is a locked door on the path, and multiple misleading locked doors in the environment. Do not worry about the key.

A reward of $1 - 0.9 * (step\_count/max\_steps)$ is given for success, and 0 for failure.

### A.4 PROGRESS FUNCTION SAMPLES

SwingCup:

```python
def progress_function(self) -> Tuple[List[torch.Tensor],
    List[bool]]:
    # Stage 1: Gripping Handles
    # Calculate distances of hands to respective handles
    dist_left_hand_to_left_handle = self.dist(self.left_hand_pos,
        self.cup_left_handle_pos)
    dist_right_hand_to_right_handle = self.dist(self.right_hand_pos,
        self.cup_right_handle_pos)
    # Average distance to handles for gripping as one progress
        metric
    avg_grip_distance = (dist_left_hand_to_left_handle +
        dist_right_hand_to_right_handle) / 2

    # Stage 2: Orienting Cup
    # Calculate rotational distance to the goal orientation
    orientation_error = self.rot_dist(self.object_rot,
        self.goal_rot)

    # Collect progress variables and directions (True for increasing,
        False for decreasing)
    progress_vars = [avg_grip_distance, orientation_error]
    progress_directions = [False, False]  # Both should decrease

    return progress_vars, progress_directions
```

CatchUnderarm

```python
def progress_function(self):
    # Calculate distance between the object's current position and
        the goal.
    object_to_goal_distance = self.goal_dist(self.object_pos)  #
        Calculated via the helper function.

    # Since the task progress is measured by how close the object
        comes to the goal,
    # we want this distance to decrease. Thus, progression in the
        task is achieved
    # when `object_to_goal_distance` is minimized.
    progress_vars = [object_to_goal_distance]
    progress_directions = [False]  # False indicates that we require
        the distance to decrease to indicate progress.

    return progress_vars, progress_directions
```

DoorCloseOutward

```python
def progress_function(self):
    # Calculate distance between the object's current position and
        the goal.
    object_to_goal_distance = self.goal_dist(self.object_pos)  #
        Calculated via the helper function.

    # Since the task progress is measured by how close the object
        comes to the goal,
    # we want this distance to decrease. Thus, progression in the
        task is achieved
    # when `object_to_goal_distance` is minimized.
    progress_vars = [object_to_goal_distance]
```

```
9       progress_directions = [False]  # False indicates that we require
        ↪   the distance to decrease to indicate progress.
10
11      return progress_vars, progress_directions
```

## A.5    ABLATIONS

**No feature library**    Failed CatchUnderarm code:

```python
def progress_function(self):
    # Calculate Euclidean distance between the current object
    ↪   position and the goal position
    object_to_goal_distance = torch.norm(self.object_pos -
    ↪   self.goal_pos, dim=-1)

    # Calculate the norm of the linear velocity of the object
    object_velocity_magnitude = torch.norm(self.object_linvel,
    ↪   dim=-1)

    # For stage 1, we want the distance to decrease
    # For stage 2, we want the velocity's magnitude to decrease
    ↪   (indicating soft catching)
    return [object_to_goal_distance, object_velocity_magnitude],
    ↪   [False, False]
```

**No heuristic discretization**    Failed SwingCup code:

```python
def progress_function_no_discretization(self) -> torch.Tensor:
    # Constants for scales or thresholds
    POSITION_THRESH = 0.05  # acceptable position error in meters
    ROTATION_THRESH = 0.1  # acceptable rotation error in radians

    # Compute distances of hands to their respective cup handles
    left_handle_distance = self.dist(self.left_hand_pos,
    ↪   self.cup_left_handle_pos)
    right_handle_distance = self.dist(self.right_hand_pos,
    ↪   self.cup_right_handle_pos)

    # Compute rotational distance to the goal orientation for the
    ↪   cup
    cup_orientation_error = self.rot_dist(self.object_rot,
    ↪   self.goal_rot)

    # Assess position accuracy
    left_hand_position_accuracy = (left_handle_distance <
    ↪   POSITION_THRESH).float()
    right_hand_position_accuracy = (right_handle_distance <
    ↪   POSITION_THRESH).float()

    # Assess rotation accuracy
    rotation_accuracy = (cup_orientation_error <
    ↪   ROTATION_THRESH).float()

    # Combine these measures into an overall progress metric (scaled
    ↪   to 0-1000)
    total_accuracy = left_hand_position_accuracy *
    ↪   right_hand_position_accuracy * rotation_accuracy
    progress_bins = (total_accuracy * 1000).long()  # Convert to
    ↪   long for discrete bin values

    return progress_bins
```

## A.6 PROGRESS DISCRETIZATION HEURISTICS

**Bi-DexHands**

Code logic: for an increasing variable, the min progress value is the value at the start of the episode. If the variable is greater than zero, the max is tracked as the max of the progress values seen so far. If the variable is less than zero, the max is set to 0. Then progress is rescaled between the min and max. For a decreasing variable, the complementary logic holds.

```python
def compute_bin_from_progress(self, progress_vars,
    progress_directions):
    """
    Compute the binning from the progress variables and directions
    """
    # First, set min/max values if not already set
    # Also normalize progress vars in here
    for i in range(len(progress_directions)):
        if progress_directions[i]:
            if 'min' + str(i) not in self.extras:
                self.extras['min' + str(i)] =
                    torch.min(progress_vars[i])
            if 'max' + str(i) in self.extras:
                self.extras['max' + str(i)] =
                    torch.max(torch.tensor([torch.max(progress_vars[i]),
                    self.extras['max' + str(i)]]))
            else:
                self.extras['max' + str(i)] =
                    torch.max(progress_vars[i])
            if self.extras['max' + str(i)] < 0:
                progress_vars[i] = torch.clamp((progress_vars[i] -
                    self.extras['min' + str(i)]) /
                    (-self.extras['min' + str(i)]), min=0, max=1)
            else:
                progress_vars[i] = torch.clamp((progress_vars[i] -
                    self.extras['min' + str(i)]) /
                    (self.extras['max' + str(i)] - self.extras['min'
                    + str(i)]), min=0, max=1)
        else:
            if 'max' + str(i) not in self.extras:
                self.extras['max' + str(i)] =
                    torch.max(progress_vars[i])
            if 'min' + str(i) in self.extras:
                self.extras['min' + str(i)] =
                    torch.min(torch.tensor([torch.min(progress_vars[i]),
                    self.extras['min' + str(i)]]))
            else:
                self.extras['min' + str(i)] =
                    torch.min(progress_vars[i])
            if self.extras['max' + str(i)] < 0:
                progress_vars[i] = torch.clamp((self.extras['max' +
                    str(i)] - progress_vars[i]), min=0) /
                    (self.extras['max' + str(i)] - self.extras['min'
                    + str(i)])
            else:
                progress_vars[i] = torch.clamp((self.extras['max' +
                    str(i)] - progress_vars[i]), min=0) /
                    self.extras['max' + str(i)]
    print("Extras", self.extras)
    # Progress is now associated with increasing values for both
    #   bins...
    # So we can generate an overall progress bin by just adding them
    #   together, with the appropriate granularity/scaling
    binning = torch.zeros(progress_vars[0].shape, dtype=torch.long,
        device=progress_vars[0].device)
```

```
34      for i in range(len(progress_vars)):
35          binning += ((progress_vars[i] * (1000 * (i ==
        ↪  (len(progress_vars) - 1)) + 20)).long() % 10000)
36      # Now generate bins from normalized vars
37      return binning
```

**MiniGrid**

```
1  def discretize_progress(self, obs, max_progress):
2      # Progress is decreasing, max_progress is set as the progress
       ↪  value at the start of the episode
3      # Get progress vars
4      progress_vars, _ = self.progress_function()
5
6      # Replace infs and nans in progress_vars
7      progress_vars = [0 if math.isnan(var) or math.isinf(var) else var
       ↪  for var in progress_vars]
8
9      # Clip by max progress
10     if max_progress is None:
11         max_progress = [elem for elem in progress_vars]
12     else:
13         progress_vars = [min(var, max_progress[i]) for i, var in
           ↪  enumerate(progress_vars)]
14
15     # Combine bins
16     obs['goal_distance'] = progress_vars[0] +
       ↪  100*progress_vars[1]*(progress_vars[0] == 0)
17
18     return obs, max_progress
```

## A.7 BI-DEXHANDS RESULTS

The results in this section correspond to Fig. 3. The bar chart numbers are reflected in tabular form. All numbers are measured as averages across trials. For ProgressCounts, we also include the standard deviation across 5 trials. For Eureka, we simply report the mean since the standard deviation is not included in the original Eureka paper.

| Task name | ProgressCounts | Eureka |
|---|---|---|
| Over | $0.93 \pm 0.01$ | 0.92 |
| DoorCloseInward | $1.00 \pm 0.00$ | 1.00 |
| DoorCloseOutward | $0.90 \pm 0.13$ | 0.96 |
| DoorOpenInward | $0.07 \pm 0.15$ | 0.00 |
| DoorOpenOutward | $0.99 \pm 0.01$ | 1.00 |
| Scissors | $1.00 \pm 0.00$ | 1.00 |
| Swing cup | $0.97 \pm 0.02$ | 0.66 |
| Switch | $0.00 \pm 0.00$ | 0.00 |
| Kettle | $0.99 \pm 0.01$ | 0.89 |
| LiftUnderarm | $0.22 \pm 0.24$ | 0.70 |
| Pen | $0.49 \pm 0.06$ | 0.57 |
| BottleCap | $0.94 \pm 0.03$ | 0.32 |
| CatchAbreast | $0.56 \pm 0.04$ | 0.50 |
| CatchOver2UnderArm | $0.90 \pm 0.03$ | 0.90 |
| CatchUnderarm | $0.76 \pm 0.05$ | 0.67 |
| ReOrientation | $0.03 \pm 0.00$ | 0.31 |
| GraspAndPlace | $0.99 \pm 0.01$ | 0.50 |
| BlockStack | $0.05 \pm 0.05$ | 0.14 |
| PushBlock | $0.03 \pm 0.03$ | 0.09 |
| TwoCatchUnderarm | $0.03 \pm 0.02$ | 0.00 |

Table 8: **A comparison of task performance between ProgressCounts and Eureka across the 20 tasks in Bi-DexHands.** Results are averaged across 5 trials for both methods, standard deviation is reported for ProgressCounts.

## A.8 PROGRESS VS PROGRESS DIFFERENCES AS REWARD

While some prior work on reward shaping uses the equivalent of progress differences as reward, we find that using progress directly as reward leads to better average task success rate than using progress differences on Bi-DexHands. Therefore, in the main paper we use progress directly as a reward when comparing to count-based rewards in Section 5.3.

| Task name | ProgressAsReward | ProgressDifferenceAsReward |
|---|---|---|
| Average | 0.45 | 0.32 |
| Over | 0.90 | 0.88 |
| DoorCloseInward | 1.00 | 0.96 |
| DoorCloseOutward | 1.00 | 0.00 |
| DoorOpenInward | 0.00 | 0.31 |
| DoorOpenOutward | 0.31 | 0.00 |
| Scissors | 1.00 | 1.00 |
| SwingCup | 0.99 | 0.94 |
| Switch | 0.00 | 0.00 |
| Kettle | 0.00 | 0.04 |
| LiftUnderarm | 0.08 | 0.35 |
| Pen | 0.22 | 0.00 |
| BottleCap | 0.04 | 0.00 |
| CatchAbreast | 0.49 | 0.01 |
| CatchOver2UnderArm | 0.94 | 0.00 |
| CatchUnderarm | 0.88 | 0.82 |
| ReOrientation | 0.06 | 0.03 |
| GraspAndPlace | 0.98 | 1.00 |
| BlockStack | 0.00 | 0.00 |
| PushBlock | 0.02 | 0.00 |
| TwoCatchUnderarm | 0.01 | 0.00 |

Table 9: **Using progress directly as reward leads to better average task success rate than using progress differences on Bi-DexHands.**

## A.9 MINIGRID TRAINING CURVES

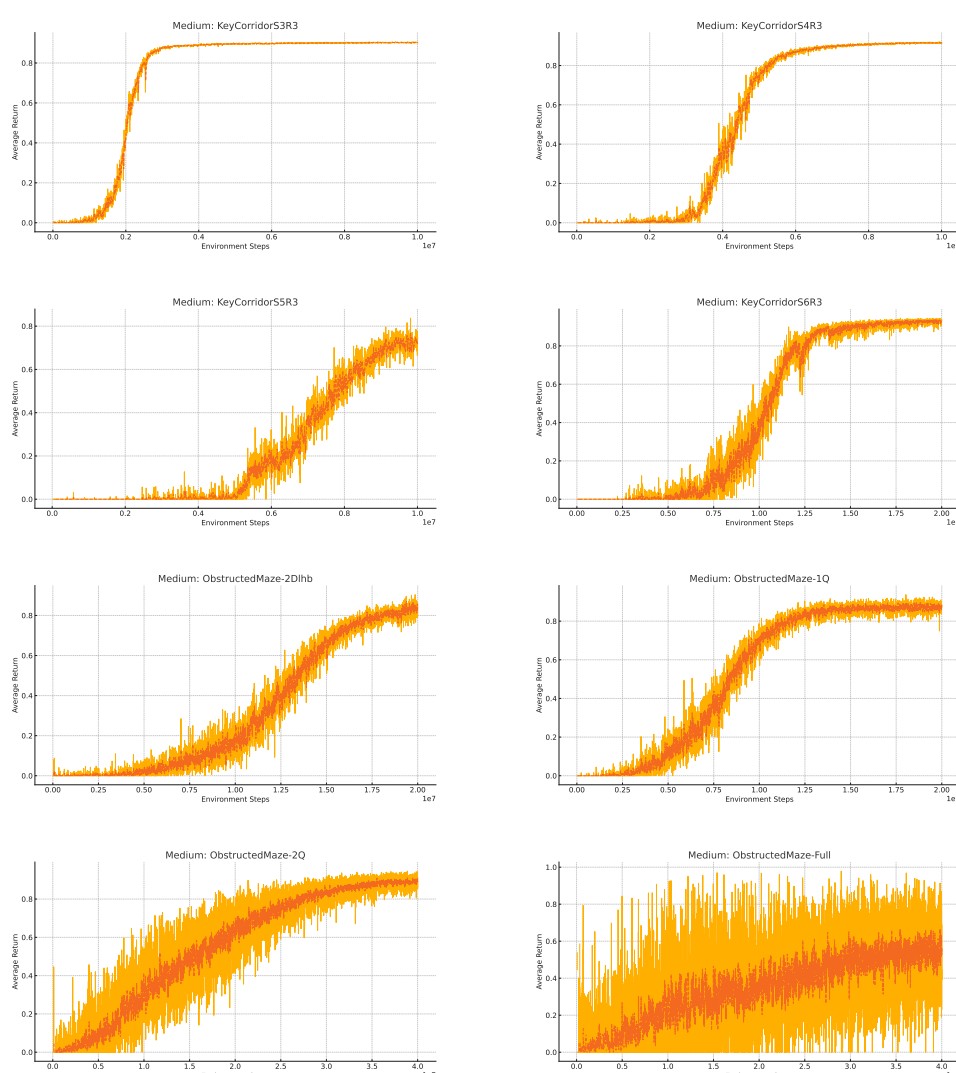

Figure 5: **Training curves for ProgressCounts on 8 hard-exploration MiniGrid tasks.**

## A.10 EXPERIMENTAL DETAILS AND HYPERPARAMETERS

**Environments**   We train our policies on two environments: Bi-DexHands (Chen et al., 2022), and MiniGrid (Chevalier-Boisvert et al., 2024).

**Code**   For the Bi-DexHands benchmark, we build upon the codebase from Eureka (Ma et al., 2023): `https://github.com/eureka-research/Eureka`.   The repo uses the RLGames implementation of PPO for training (Makoviichuk & Makoviychuk, 2021). For the MiniGrid benchmark, we build upon the codebase from NovelD (Zhang et al., 2021): `https://github.com/tianjunz/NovelD`.   Our experimental code is is available at the following anonymous link: `https://drive.google.com/drive/folders/1G88Je0K4BuexWhE8ZLgoM0WM6vHcBrvt?usp=sharing`.

**Hyperparameters**   For all Bi-DexHands tasks, we scale extrinsic rewards by $0.05$, and normalize intrinsic rewards to a mean of $0.001$. Elsewhere, we use the default hyperparameters associated with Eureka, which are the default parameters from the original Bi-DexHands benchmark. Progress functions are discretized into $1020$ bins for count-based exploration (for tasks with two subtasks, the first subtask is discretized to $20$ bins, and the second subtask to $1000$ bins). For all MiniGrid tasks, we set an intrinsic reward coefficint of $0.5$, and leave all other hyperparameters at default values. Progress functions are discretized into $50$ bins for count-based exploration (for tasks with two subtasks, each subtask is discretized to $25$ bins).

**Compute**   All experiments were run on a machine with 8 NVidia Tesla V100 GPUs across 3 weeks.