# OpenReview forum: "Automated Rewards via LLM-Generated Progress Functions"
_ICLR.cc/2025/Conference — Submitted to ICLR 2025_

### Official Review · Reviewer_7kFc · 2024-10-29

**Soundness:** 2
**Presentation:** 2
**Contribution:** 2
**Rating:** 5
**Confidence:** 3

**Summary:**

The paper presents an innovative framework that uses LLMs to generate task-specific progress functions for reinforcement learning. The proposed system aims to overcome the challenge of reward engineering in sparse-reward tasks by focusing on generating progress-based intrinsic rewards. The paper introduces a two-step approach that uses LLMs to generate progress functions and then applies count-based intrinsic rewards. The authors evaluate their approach, ProgressCounts, on the challenging Bi-DexHands benchmark, and show that it achieves comparable or better results than previous state-of-the-art methods while requiring significantly fewer samples.

**Strengths:**

- The idea of simplifying reward generation by reducing it to the problem of estimating task progress is innovative. Leveraging LLMs to generate task-specific progress functions is a creative application that shows promise in reducing reliance on dense reward shaping.

- The method shows significant sample efficiency gains compared to Eureka, requiring up to 20 times fewer reward function samples while achieving state-of-the-art performance on the Bi-DexHands benchmark. This demonstrates the potential of the approach in reducing the computational cost of training.

- The experimental results on Bi-DexHands are impressive, as ProgressCounts outperforms both manually designed dense reward functions and existing automated reward generation methods in terms of task success rates.

**Weaknesses:**

- The evaluation is conducted solely on synthetic benchmarks like Bi-DexHands and MiniGrid, which may not be fully representative of real-world environments. The generalizability of ProgressCounts to more diverse and real-world complex environments remains uncertain.

- The method relies heavily on LLMs to generate progress functions, which introduces potential biases inherent to LLMs. These biases can affect the quality and reliability of progress estimates, particularly in environments where the LLM lacks specific domain knowledge or makes incorrect inferences.

- The need for a pre-defined feature engineering library for each domain limits the method's automation potential. Although the feature library is relatively simple to create, this step still requires manual intervention and domain-specific expertise, which may limit scalability and practical deployment.

- While the proposed count-based intrinsic reward framework is well-developed, the paper lacks a thorough exploration or comparison with other recent reward design techniques, such as curiosity-driven approaches or model-based reward mechanisms. This leaves open the question of whether ProgressCounts is the optimal method for all types of tasks.

- The paper does not address the scalability of ProgressCounts when moving to tasks with highly complex environments or large state-action spaces. Specifically, the discretization strategy and count-based approach may face challenges when scaling to significantly more complex scenarios, potentially reducing sample efficiency.

**Questions:**

How well does the proposed ProgressCounts framework generalize to more complex, real-world environments beyond synthetic benchmarks like Bi-DexHands and MiniGrid? Have any tests been conducted to validate its applicability to a broader set of domains?

Given that the progress functions are generated using LLMs, how does the framework address potential biases that the LLM may introduce, particularly in environments where LLMs may not have specific domain knowledge?

The paper relies on a discretization strategy and count-based intrinsic rewards. How scalable is this approach when applied to tasks with significantly larger state-action spaces or more complex environments? Would sample efficiency gains still hold under such conditions?

---

> ### Author Response · Authors · 2024-11-23
>
> Thank you for your thoughtful and detailed feedback! We appreciate that you see our approach as a simplification of reward generation, and you find our experimental results effective.
>
> > The evaluation is conducted solely on synthetic benchmarks like Bi-DexHands and MiniGrid, which may not be fully representative of real-world environments. The generalizability of ProgressCounts to more diverse and real-world complex environments remains uncertain.
>
> We agree that our evaluation is restricted solely to synthetic benchmarks–we test on the sparse-reward benchmarks tested in Eureka (Bi-DexHands), and we also offer an additional benchmark (MiniGrid). This is common practice across a wide range of reinforcement learning work in the community. We agree that future work should explore the transfer of ProgressCounts-based policies to real-world tasks by integrating ProgressCounts with sim-to-real research.
>
> > The method relies heavily on LLMs to generate progress functions, which introduces potential biases inherent to LLMs. These biases can affect the quality and reliability of progress estimates, particularly in environments where the LLM lacks specific domain knowledge or makes incorrect inferences.
>
> We note that our framework is compatible with techniques to mitigate biases, such as retrieval-augmented generation (RAG) or the use of guardrails (e.g., NVIDIA NeMo-Guardrails). Moreover, unlike human designers, LLMs can be systematically augmented or constrained, potentially resulting in less bias over time.
>
> > The need for a pre-defined feature engineering library for each domain limits the method's automation potential. Although the feature library is relatively simple to create, this step still requires manual intervention and domain-specific expertise, which may limit scalability and practical deployment.
>
> Please see our note on creating a feature engineering library in the general response.
>
> > While the proposed count-based intrinsic reward framework is well-developed, the paper lacks a thorough exploration or comparison with other recent reward design techniques, such as curiosity-driven approaches or model-based reward mechanisms. This leaves open the question of whether ProgressCounts is the optimal method for all types of tasks.
>
> Our paper contributes the progress function abstraction for extracting task-specific domain knowledge from the LLM, and also proposes count-based rewards as a simple, effective mechanism to leverage progress functions to provide rewards. We do not necessarily believe that ProgressCounts is optimal for all types of tasks–we offer the observation that count-based rewards are a promising alternative to dense reward shaping, and leave the exploration of other reward mechanisms to future work.
>
> > The paper does not address the scalability of ProgressCounts when moving to tasks with highly complex environments or large state-action spaces. Specifically, the discretization strategy and count-based approach may face challenges when scaling to significantly more complex scenarios, potentially reducing sample efficiency.
>
> As environments and state-action spaces vary in complexity, the progress function’s role is to reduce the complexity to simple scalar features, which are then converted into scalar rewards. Even as the complexity of the environment and state-action space increase, because discretization happens at the progress level, not the state-space level, the complexity of the inputs to the discretization and count-based components remains the same.

---

> > ### Comment · Reviewer_7kFc · 2024-11-24
> >
> > Thanks for your kind response. I would prefer to retain my original scores.

---

### Official Review · Reviewer_mw3c · 2024-11-04

**Soundness:** 3
**Presentation:** 3
**Contribution:** 2
**Rating:** 5
**Confidence:** 3

**Summary:**

This submission concerns automatic reward shaping for RL tasks by utilizing LLMs for code synthesis, i.e., generating a shaping function. In particular, it improves on prior work (EUREKA) in constraining LLM output to "progress functions", which can serve as a basis for state binning, which in turn enables the use of count-based intrinsic rewards. Putting everything together, the authors demonstrate strong performance on a robot hand manipulation benchmark while significantly reducing the required samples of EUREKA (which generates variations of shaping functions by re-prompting the LLM with feedback obtained from running an RL training algorithm with the shaping function).

**Strengths:**

For the most part, the paper is easy to read and follow; the motivation is clear and the method is described well. Generally, I find the link between automatically generating a symbolic notion of task progress by an LLM (task description to progress function) and utilizing it to derive count-based rewards (which can be very effective if designed well) original and interesting; the experimental results demonstrate the efficacy quite well. I also liked that the evaluation focused on required training runs and environment interactions, where the proposed method is vastly superior to EUREKA.

**Weaknesses:**

First, I have doubts regarding the significance of this work. The main message -- you can get more out of an LLM's ability to translate language to RL specifications by placing additional constraints and supplying further human knowledge -- is timely and line with other recent works from the code generation community. On the other hand, the paper largely follows the problem setting of EUREKA, which, to my knowledge, has yet to demonstrate value beyond being an interesting application of LLMs. The original paper on defining rewards with LLMs (https://arxiv.org/abs/2306.08647) utilized model predictive control to directly translate the reward specifications to actions; count-based exploration does not allow for this, however, and relies on training a policy over millions of time-steps.

Further points:
- In Table 1, the same number of trials should be used for all methods. From 5.1. I understand that you select the best trial run, so a fair comparison should provide the same number of trials to all methods considered.

Finally,  the presentation could benefit from additional clarity, in particular the experimental section is at times hard to follow. Some concrete suggestions:
- In the first sentence of the abstract, you might want to mention that you're ultimately concerned with RL. "Automated reward engineering" is not a term I could immediately place in a specific field.
- Please provide the `Env` class. It's not clear to me whether the progress function has access to the same state as the policy or whether additional variables are provided
- Supply hyper-parameters such as learning rate for PPO
- The claim that the feature engineering library can be authored in minute (L164) is quite bold, in particular considering new domains and a lack of guidance for non-RL experts on what this library should contain.
- In Figure 3 or in the corresponding text, be explicit on how many environment samples are used.

**Questions:**

- Do your progress function break the MDP assumption? E.g., the BFS helper method for MiniGrid requires full access to the full environment state?
- Did you try prompting the LLM to always supply progress variables where progress strictly means an increase in value? This way you would not need the additional variables.

These questions should be answered in the text:
- For the MiniGrid experiment, how many runs of ProgressCounts were done?
- On which data do you base the range estimation of the progress variables? I understand you need several example rollouts (ideally successful) for this?

---

> ### Author Response · Authors · 2024-11-23
>
> Thank you for your thoughtful and thorough feedback! We appreciate that you find our method original and interesting, and you appreciate the importance of our gains in sample efficiency.
>
> > In Table 1, the same number of trials should be used for all methods. From 5.1. I understand that you select the best trial run, so a fair comparison should provide the same number of trials to all methods considered.
>
> To clarify, there is a distinction between the number of training trial runs (from which we select the best trial run) and the number of evaluation runs (which are used to measure the performance of the best trial run). In Table 1, we ran a single evaluation run for the methods considered due to computational constraints–which does increase the variance of the estimate, but we do NOT select between runs in evaluation. In the final version of the paper, given more time to run trials, we will run 5 evaluation trials for all methods. Please see the general response for a clarification of the number of trials for training vs evaluation.
>
> > In the first sentence of the abstract, you might want to mention that you're ultimately concerned with RL. "Automated reward engineering" is not a term I could immediately place in a specific field.
>
> Thank you, we now mention this.
>
> > Please provide the Env class. It's not clear to me whether the progress function has access to the same state as the policy or whether additional variables are provided.
>
> Each of the 20 Bi-DexHands tasks has a different Env class. The progress function has access to the same state space as a reward function–following an apples-to-apples comparison with Eureka. There are no additional variables beyond those provided to the Eureka reward functions. We clarify this in lines 298-299.
>
> > The claim that the feature engineering library can be authored in minutes (L164) is quite bold, in particular considering new domains and a lack of guidance for non-RL experts on what this library should contain.
>
> We have refined the claim to clarify that we are referring to RL experts crafting the type of feature engineering library that would also be valuable for dense reward shaping.
>
> > In Figure 3 or in the corresponding text, be explicit on how many environment samples are used. Supply hyper-parameters such as learning rate for PPO.
>
> Following the protocol from Eureka, we have adopted the same number of environment samples (100M) and the same PPO hyperparameters (the ones provided by the original Bi-DexHands paper). We clarify this in lines 307-309. We **emphasize that our setup matches Eureka unless otherwise noted: we borrow all PPO hyperparameters from Bi-DexHands; these parameters are optimized for human-written dense rewards.** All Bi-DexHands policies are trained on 100 million samples, identical to the prior work. Additionally, note that the progress function has access to the same state variables as dense rewards (Lines 297-98), ensuring a fair comparison with reward shaping methods.
>
> > Do your progress function break the MDP assumption? E.g., the BFS helper method for MiniGrid requires full access to the full environment state?
>
> The progress function, taking the place of a dense reward function, has access to the same information as a reward function: the environment state. This does not break the MDP assumption since the progress function is simply part of the mechanism to provide rewards to learning.
>
> > Did you try prompting the LLM to always supply progress variables where progress strictly means an increase in value? This way you would not need the additional variables.
>
> This is a good idea that we have not yet tried–we will try this in future extensions to the work to simplify the progress formulation.
>
> > For the MiniGrid experiment, how many runs of ProgressCounts were done?
>
> We run 4 trials of reward function generation, as we do with Bi-DexHands. This has been updated in Lines 666-667.
>
> > On which data do you base the range estimation of the progress variables? I understand you need several example rollouts (ideally successful) for this?
>
> For increasing progress variables, the minimum is initialized as the first observed value, and the maximum is tracked dynamically–bins are normalized between the minimum value and the maximum value seen so far. Complementary logic applies to decreasing variables. We have updated our pseudocode in Appendix A.6 along with a description of the logic for greater clarity.

---

### Official Review · Reviewer_foaV · 2024-11-08

**Soundness:** 3
**Presentation:** 3
**Contribution:** 3
**Rating:** 8
**Confidence:** 2

**Summary:**

This work proposes an algorithm to generate progress functions with fewer samples. This is accomplished by mapping the original observation space to a lower-dimensional representation, discretizing this space, and calculating intrinsic rewards based on the inverse square root of visitation counts.

Compared to previous baselines, this approach achieves state-of-the-art (SOTA) performance while utilizing fewer samples.

**Strengths:**

- The use of domain-specific discretization within the projected subtask progression space is novel and yields promising results.
- The experiments are thorough, with ablation studies providing valuable insights into the function of each design component.
- The empirical results are robust, demonstrating improved performance alongside reduced sample complexity.

**Weaknesses:**

In Figure 4, could you also provide results from Eureka? It would be interesting to see how the proposed method and Eureka performance improve as more samples are available.

**Questions:**

How are the additional variables $[y_1, y_2, \ldots, y_k]$ used?

---

> ### Author Response · Authors · 2024-11-23
>
> Thank you for your feedback! We appreciate that you see the promise of domain-specific discretization, and agree that our empirical results are thorough and robust.
>
> > In Figure 4, could you also provide results from Eureka? It would be interesting to see how the proposed method and Eureka performance improve as more samples are available.
>
> Please note that Figure 4 is already an iso-sample experiment: by allocating 20x the samples to a single training run, we leverage the same sample budget in 4 training runs that Eureka leverages across 80 training runs. We attempted to re-run the Eureka codebase to test with even more samples, but we were unable to replicate their results–we only achieved a task success rate of 0.32 under the original experimental conditions, whereas their original reported results had a task success rate of 0.56.
>
> > How are the additional variables y_i used?
>
> The variables y_i inform us whether the progress is an increasing or decreasing quantity. This helps estimate the minimum and maximum values progress should take: for an increasing variable, the minimum value of progress is the value at the first timestep, and for a decreasing variable, the maximum value of progress is the value at the first timestep. For further intuition code establishing the heuristics normalizing and discretizing progress, please see Appendix A.6.

---

### Official Review · Reviewer_euqU · 2024-11-10

**Soundness:** 2
**Presentation:** 3
**Contribution:** 2
**Rating:** 5
**Confidence:** 3

**Summary:**

The paper presents a novel framework leveraging Large Language Models (LLMs) to automate the generation of reward functions for reinforcement learning tasks. The authors claim significant improvements in sample efficiency and performance over prior state-of-the-art methods on the Bi-DexHands benchmark.

**Strengths:**

1. The paper is well-organized and easy to follow, making complex concepts accessible.
2. The overall motivation behind using LLMs for automated reward engineering is compelling.
3. The experimental results are substantial, demonstrating clear performance gains over existing methods.

**Weaknesses:**

1. The motivation for mapping progress to bins as a representation for intrinsic rewards is not clearly articulated. A common principle in intrinsic reward design is that the representation space should effectively capture the essential aspects of the original observation space. By incrementing counters in this space, the algorithm should be able to explore the full state space effectively. The authors need to provide a clearer rationale for why progress bins are chosen over other potential representations.
2. The method for selecting the best progress function among the generated options is not detailed (Line 287-289). The authors should provide a clear selection criterion to ensure reproducibility and allow for comparison of different progress functions.
3. The paper claims that progress-based bins are superior to simhash, yet it is not evident why this should be the case. Simhash is known to provide convergence guarantees by ensuring that increasing intrinsic rewards correspond to increased visits to the original full state space, though potentially requiring longer training times.
4. On Line 427, the authors suggest using the cumulative progress as a step reward for task completion, which is misguided. The correct approach, as per reward shaping principles, is to use the difference in progress.

**Questions:**

see weaknesses above.

---

> ### Author Response · Authors · 2024-11-23
>
> Thank you for your thorough and detailed feedback! We appreciate that you see our significant gains in task success rate and, in particular, sample efficiency compared to prior work.
>
> > The motivation for mapping progress to bins as a representation for intrinsic rewards is not clearly articulated. A common principle in intrinsic reward design is that the representation space should effectively capture the essential aspects of the original observation space. By incrementing counters in this space, the algorithm should be able to explore the full state space effectively. The authors need to provide a clearer rationale for why progress bins are chosen over other potential representations.
>
> > The paper claims that progress-based bins are superior to simhash, yet it is not evident why this should be the case. Simhash is known to provide convergence guarantees by ensuring that increasing intrinsic rewards correspond to increased visits to the original full state space, though potentially requiring longer training times.
>
> Prior work on SimHash-based counts establishes that, while lacking the same empirical guarantees, count-based reward perform better alongside human-written hash functions than with generic hash functions–human domain knowledge allows for the selection of relevant features in the hash function, driving exploration along key axes of the state space. LLM-generated progress functions play a similar role by leveraging the LLM’s domain knowledge. Please see the section of the general response “Why progress functions over SimHash” for a more extended discussion.
>
> > The method for selecting the best progress function among the generated options is not detailed (Line 287-289). The authors should provide a clear selection criterion to ensure reproducibility and allow for comparison of different progress functions.
>
> We select the progress function with the best task success rate from a single run–the same experimental protocol as Eureka. This selection criterion is in Lines 311-313, and also in the original PDF. Please see the “How we select the best progress function, and how we evaluate it” section in the general response.
>
> > On Line 427, the authors suggest using the cumulative progress as a step reward for task completion, which is misguided. The correct approach, as per reward shaping principles, is to use the difference in progress.
>
> We acknowledge that using cumulative progress as a reward may be less conventional than using progress differences. However, on Bi-DexHands we ran experiments using both approaches, and **cumulative progress achieved an average success rate of 0.45, whereas using progress differences only achieved an average success rate of 0.32**. Therefore, we included results directly using progress as reward to offer the best possible comparison to dense rewards. We have presented the difference between these two dense reward approaches in **our updated Appendix A.8.**

---

> ### Comment · Reviewer_euqU · 2024-12-01
>
> Thanks for the rebuttal. I believe that the progress function can be good guidance for exploration, but, rigorously speaking, I do not agree that it can be connected with count-based rewards, which is used to guide the agent to cover state space equally. I prefer to retain my original scores.

---

> > ### Author Response · Authors · 2024-12-02
> >
> > We are glad you agree that progress functions provide good guidance for exploration.
> >
> > We believe that our application of progress function-based bins with count-based rewards is well-motivated because our algorithm follows two standard steps, **both supported by precedent in prior work**:
> >
> > 1. Define a mechanism to track state visitation counts. Given a high-dimensional state space, reducing the state space to a lower-dimensional space makes count-based rewards more tractable and efficient. This can be done by defining a hashing function: SimHash [1], a human-defined function [1, 2], or in our case, a discretization via the LLM-defined progress function. Alternatively, state pseudo-counts [3] can be tracked via a learned density model [3, 4].
> > 2. Use the state visitation counts to apply count-based rewards [5, 6].
> >
> > **In the interest of improving our method, we would appreciate clarification on where either step lacks rigor.**
> >
> > ##### [1] Tang, Haoran, et al. "# exploration: A study of count-based exploration for deep reinforcement learning." Advances in neural information processing systems 30 (2017).
> >
> > ##### [2] Ecoffet, Adrien, et al. "First return, then explore." Nature590.7847 (2021): 580-586.
> >
> > ##### [3] Bellemare, Marc, et al. "Unifying count-based exploration and intrinsic motivation." Advances in neural information processing systems 29 (2016).
> >
> > ##### [4] Ostrovski, Georg, et al. "Count-based exploration with neural density models." International conference on machine learning. PMLR, 2017.
> >
> > ##### [5] Strehl, Alexander L., and Michael L. Littman. "An analysis of model-based interval estimation for Markov decision processes." Journal of Computer and System Sciences 74.8 (2008): 1309-1331.
> >
> > ##### [6] Kolter, J. Zico, and Andrew Y. Ng. "Near-Bayesian exploration in polynomial time." Proceedings of the 26th annual international conference on machine learning. 2009.

---

### Author Response · Authors · 2024-11-23
**Response to reviewer feedback**

## Summary of reviews

We thank the reviewers for their insightful feedback and constructive suggestions. Our work presents an innovative framework for generating reward functions by reducing reward engineering to task progress estimation and leveraging LLMs to generate progress functions. This approach demonstrates substantial **sample efficiency gains**, requiring up to 20x fewer reward function samples than Eureka (ICLR 2024) while achieving state-of-the-art performance on Bi-DexHands (foaV, 7kFc). We provide **strong empirical results** on both Bi-DexHands and MiniGrid benchmarks, illustrating both superior performance and reduced computational costs (foaV, mw3c). Furthermore, we conducted **thorough ablation studies** to isolate and evaluate the contributions of individual design components, offering insights into their roles (foaV). By linking symbolic notions of task progress with count-based intrinsic rewards, our work contributes a novel method to the RL literature, which we validated with robust evaluations focusing on training efficiency and task performance (mw3c).

## Clarifications on some key points

**Focus on comparison to Eureka:** Automated reward generation has the potential to vastly reduce the amount of human iteration required for training RL policies. **We present our work in comparison to Eureka (ICLR 2024) both because it was the prior SOTA work on Bi-DexHands and because it used LLM-based reward generation.** Bi-DexHands presents a substantial state-action space comparable to real-world robotics tasks. Both progress functions and traditional reward shaping rely on simplifying the complexity of state-action spaces, and our approach demonstrates better performance in these settings.

**Creating a feature library:** We agree that creating a feature engineering library for new environments requires some human effort. However, this effort is modest compared to designing dense rewards or crafting viable observation spaces, which are typically required in RL. For practitioners already skilled in RL, we believe this tradeoff is beneficial–especially due to the SOTA performance and 20x efficiency gains. **We have revised the text to better align this claim with the expectations for RL practitioners.**

**The choice of progress functions over SimHash:** We appreciate the request for greater justification of our choice of progress bins over alternatives like SimHash. Progress bins discretize the state space by leveraging domain knowledge, which empirically allows them to better capture task-relevant aspects of the environment. As seen in prior work [1], **human-informed hash functions using domain knowledge, while lacking better theoretical properties, empirically outperform generic representations like SimHash.** SimHash offers convergence guarantees, but progress bins provide practical benefits under finite training budgets, as evidenced by our empirical results in Table 1.

**How we select the best progress function, and how we evaluate it:** Following Eureka’s experimental protocol, the progress function that achieves the highest task success during a single training run is selected **in an automated fashion (Lines 311-313, also in the original pdf)** and the selected function is evaluated over five additional runs with different seeds **(Lines 298-300, also in the original pdf). Please note the distinction between the function selection protocol (select best training run) and evaluation protocol (report the average performance of the selected function on 5 additional runs). This is the protocol from Eureka.** In the ablation presented in Table 1, the comparison is fair because the ablated methods simply are averaging across fewer evaluation trials.

**Progress vs progress differences as reward:** We acknowledge that using cumulative progress as a reward may be less conventional than using progress differences. However, on Bi-DexHands we ran experiments using both approaches, and cumulative progress achieved an average success rate of 0.45, whereas using progress differences only achieved an average success rate of 0.32. Therefore, we included results directly using progress as reward to offer the best possible comparison to dense rewards. **We have presented the difference between these two dense reward approaches in our updated Appendix A.8.**

[1] Haoran Tang, Rein Houthooft, Davis Foote, Adam Stooke, OpenAI Xi Chen, Yan Duan, John Schulman, Filip DeTurck, and Pieter Abbeel. # exploration: A study of count-based exploration for deep reinforcement learning. Advances in neural information processing systems, 30, 2017.

---

### Meta-Review · Area_Chair_egrC · 2024-12-22

**Metareview:**

This paper proposes a novel strategy for reward synthesis that leverages LLMs to generate progress functions, which can be used to discretize the state space and produce count-based rewards. While the paper exhibits gains compared to Eureka, the main baseline, these gains are somewhat modest. In addition, it has a number of drawbacks compared to Eureka: (i) it is a complicated system that has many moving parts, and (ii) since it does not generate free-form rewards, it is not as flexible as Eureka. While the authors emphasize the reduction in the number of samples, this was not a goal of Eureka, and there are likely simpler ways to improve Eureka along this dimension.

**Additional Comments On Reviewer Discussion:**

Reviewers raised concerns about the ad-hoc nature of the framework, limitations in the experiments, as well as the incremental nature compared to Eureka. While some of these concerns were addressed during the rebuttal period, the reviewers who engaged with the authors remained unconvinced.

---

### Decision · Program_Chairs · 2025-01-22

Reject